# LOCAL DISTRIBUTION-CONDITIONED IMAGE SYNTHESIS FOR ONE-SHOT FEDERATED LEARNING

## ABSTRACT

One-Shot Federated Learning (OSFL) aims to build a global model with a single round of server–client interaction, making it attractive for practical scenarios. The recent introduction of Diffusion Models has enabled OSFL to synthesize client-like data on the server. However, these methods typically require fine-tuning a foundation model or a shared feature extractor on clients, which undermines practicality under the heterogeneous scenarios. To address this limitation, we propose **FedLD**, a one-shot **Fed**erated learning method with **L**ocal **D**istribution-conditioned image synthesis. We fit a Gaussian Mixture Model (GMM) to the local distribution of each client and upload the model parameters to the server. The server samples initial noise from these client-specific models to guide a Diffusion Model to generate data aligned with the client distributions, enabling OSFL without any client-side model training and significantly reducing both computation and communication costs. Quantitation and visualization experiments conducted on three large-scale real-world image datasets demonstrate the initial noises sampled from the GMMs can effectively transfer the knowledge of client distributions, further validating the potential of Diffusion Models in OSFL.

## 1 INTRODUCTION

One-Shot Federated Learning (OSFL) has emerged as a promising paradigm in federated learning, aiming to build a global model through a single round of server–client interaction. Unlike traditional federated learning, which requires multiple iterations of communication and local training (McMahan et al., 2017; Lin et al., 2020; Li et al., 2020), OSFL significantly reduces client-side costs, making it more feasible in bandwidth-constrained or resource-limited settings. However, the practicality of OSFL is limited by the challenge of handling heterogeneous data distributions across clients: with only one communication round, the server lacks sufficient information to faithfully capture the diversity of client data. This motivates the need for new techniques that can effectively transfer distributional knowledge from clients to the server without imposing heavy computational or communication burdens.

A recent line of work explores leveraging Diffusion Models (DMs) to address this challenge (Yang et al., 2023; 2024; Zhang et al., 2023). With the assistance of a pre-trained conditional DM (Rombach et al., 2022), the server can generate a synthetic dataset that complies with the client's data distribution by using information from the client as conditions. This enables the training of an aggregated model that adapts to the distribution of all clients. Compared to traditional OSFL methods, DM-based methods substantially reduce the communication and computational costs of clients and can be applied to large-scale real-world images with various distributions. However, these methods typically require fine-tuning the foundation model or a unified feature extractor on the clients, which limits their practicality in heterogeneous scenarios. The underlying cause of this problem is that these methods rely on cross-attention for conditional generation, making the conditions highly coupled with the parameters of the DM, thereby requiring specialized training. Therefore, to further enhance the practicality of DM-based OSFL, new methods for conditional generation are needed.

One promising direction lies in exploiting initial noise as a conditioning signal. As the stochastic input to the diffusion process, initial noise inherently shapes the diversity and fidelity of generated images and thus offers unique potential for guiding generation. Although some works (Mao et al., 2023; Guo et al., 2024) have explored the value of initial noise in specifying object positions and

accelerating the generation process, how to effectively use initial noise for conditional generation and specify the distribution of the generated image remains an under-explored direction. Investigating the role of initial noise in conditional generation with DMs has the potential to introduce new control dimensions into the generation process, thereby expanding the boundaries of DMs' applications in both generative tasks and OSFL.

Therefore, to explore the potential application of initial noise in OSFL, we propose **FedLD**, a one-shot **Fed**erated learning method with **L**ocal **D**istribution-conditioned image synthesis, which explores the use of initial noise to guide the DM for conditional generation under the OSFL framework. Specifically, FedLD consists of two stages: local distribution fitting and server image generation. During the local distribution fitting, we use a Gaussian Mixture Model (Reynolds et al., 2009) (GMM) to learn the distribution of the noised client images, enabling the GMM to carry the client's distribution information, with only statistics of the fitted GMM being uploaded to the server. In the server image generation, the server utilizes the noises sampled from the fitted GMM as the initial noise of the diffusion process and generates the synthetic dataset that complies with the client distributions to train the aggregated model. By using initial noise as guidance for the conditional generation, FedLD provides more precise control over the style, background, and overall layout of local distributions. Additionally, since only a small amount of statistical information from the noised local distribution needs to be uploaded, FedLD incurs low computational and communication costs on the clients, further enhancing its practicality.

To evaluate the performance of FedLD, we conduct thorough quantitation and visualization experiments on three large-scale real-world image datasets: DomainNet, OpenImage, and NICO++. The experimental results demonstrate that using the client distribution information provided by the sampled initial noise from the fitted GMMs, the DM can effectively generate synthetic datasets that closely resemble the client's data style and background on the server. Additionally, compared to other cross-attention-based conditional generative methods, FedLD has a significant advantage in generating images with specific backgrounds and styles. These results validate the potential of initial noise in conditional DMs and further reveal the possibilities of applying DMs within the OSFL framework.

Our contributions are summarized as follows:

- We propose FedLD, an OSFL method with local distribution-conditioned image synthesis. Based on initial noise sampled from the fitted local distribution, FedLD enables the server to generate a synthetic dataset that complies with client distributions with highly reduced computation and communication costs on the clients, further revealing the application potential of DMs in OSFL.
- We explore the potential of initial noise in guiding the DM generation process. By adjusting the initial noise of the DM, FedLD can control the style, background, and semantics of the generated images, providing a new method for conditional generation with DMs.
- We conduct comprehensive quantitation and visualization experiments, demonstrating that, compared with traditional cross-attention-based conditional generation methods, initial noise-conditioned generation is more effective in guiding the generation of specific backgrounds and styles, effectively demonstrating the potential of initial noise in guiding conditional generation.

## 2 RELATED WORKS

### 2.1 ONE-SHOT FEDERATED LEARNING

In traditional federated learning methods, the convergence of the aggregation model requires multiple rounds of communication between the clients and the server, which results in an exponential increase in communication costs. Therefore, one-shot federated learning (OSFL) has started to gain widespread attention. OSFL requires all clients to complete local model training in a single training process and aggregate their updates to the server, allowing the global model to be updated with only one round of communication. (Guha et al., 2019; Lin et al., 2020; Li et al., 2021) train student models on the client side via knowledge distillation and upload them to the server. The server performs knowledge distillation on the received models to obtain the final aggregated model. Methods such as

(Zhou et al., 2020) eliminate the need to transmit model parameters. Instead, they perform dataset distillation (Wang et al., 2018) on the clients, transmitting relevant information from the client datasets directly to the server. (Zhang et al., 2022a; Heinbaugh et al., 2023) require training generators on the clients to preserve information about the client distributions, which are then uploaded to the server for pseudo-sample generation. (Yang et al., 2023; 2024; Zhang et al., 2023) introduce DMs into the federated learning, transmitting client information through image features, client distribution descriptions, or text prompts to guide the server's DM in conditional generation via cross-attention. These methods require the use of a unified foundation feature extractor or training based on foundation models, leading to significant communication and computational costs. In this paper, we explore the potential of initial noise-guided conditional generation in OSFL, effectively reducing the additional costs introduced by the use of DMs.

## 2.2 DIFFUSION MODELS

A notable feature of DMs is their conditional generation ability. Given appropriate conditions as guidance, such as images, text, or gradients of loss functions, DMs can generate images that comply with almost any distribution encountered in everyday life. The generated images exhibit remarkable quality and diversity. The main methods for guiding DMs in conditional generation can be divided into three categories: 1). Gradient Descent-based Guidance (Dhariwal & Nichol, 2021; Feng et al., 2022; Xie et al., 2023): These methods typically input the generated results into an auxiliary model, compute the loss function, and adjust the generation process through gradient descent. 2). Cross-Attention-based Methods (Saharia et al., 2022; Kim et al., 2022; Zhang & Agrawala, 2023): These methods, currently the mainstream, extract features from the input conditions and integrate them into the intermediate layers of the DM using cross-attention, thus influencing the generated results. 3). Initial Noise-based Methods: This category is still in its early stages. (Mao et al., 2023) manipulates the initial noise in the attention region to specify the layout in the generated result. (Guo et al., 2024) adjusts the initial noise to accelerate the generation process. This paper uses initial noise as guidance to generate images that specify client styles, backgrounds, semantics, and other information, further exploring the application potential of initial noise in conditional generation with DMs.

## 3 METHOD

We clarify the problem setting and symbol definitions, and provide a detailed description of FedLD in this section, including two main parts: Local Distribution Fitting and Server Image Generation. The overall framework of FedLD is shown in Figure 1.

### 3.1 PROBLEM SETTING

Consider an OSFL framework with $M$ clients. For client $m$, the client possesses its local dataset $\mathbf{D}_m = \{(x_i)\}_{i=1}^{R}$, which follows a unique data distribution $p_m(\mathbf{x})$. Suppose there are a total of $C$ categories, and $\mathcal{C}_m$ represents the set of possible class labels for each client and is a subset of $\{1, ..., C\}$. The server's goal is to obtain an aggregated model $\mathcal{F}_g$ that adapts to the data distributions $p_m(\mathbf{x}), m \in \{1, ..., M\}$ without directly accessing the local datasets $\mathbf{D}_m(\mathbf{x}), m \in \{1, ..., M\}$. This is achieved through a single communication round from client to server. The objective function for training the aggregated model in this framework is as follows:

$$\min_{\mathcal{F}_g \in \mathbb{R}^d} \frac{1}{M} \sum_{m=1}^{M} \mathbb{E}_{\mathbf{x} \sim \mathbf{D}_m} \left[ \ell_m(\mathcal{F}_g; \mathbf{x}) \right] \tag{1}$$

where $\mathbf{D}_m(\mathbf{x}) \sim p_m(\mathbf{x}), m \in \{1, ..., M\}$, and $\ell_m$ is the local objective function for client $m$.

From this objective function, it is clear that the aggregated model adapts to the data distribution of all clients. To achieve this, we generate a synthetic dataset $\hat{\mathbf{D}}_m(\mathbf{x}), m \in \{1, ..., M\}$ on the server that matches the data distribution of each client and use these synthetic datasets to train the global model. In the subsequent experimental section, the performance of this method will primarily be evaluated by the average classification accuracy on all local test datasets.

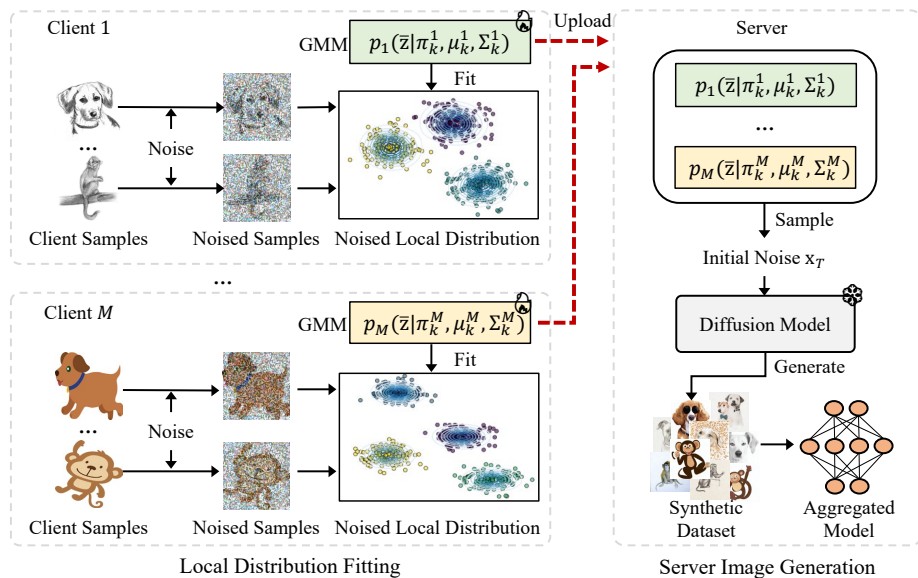

Figure 1: The overall framework of FedLD, including two main parts: Local Distribution Fitting and Server Image Generation. Firstly, we fit the local distribution of the client's noised samples using a Gaussian Mixture Model (GMM) and upload the fitted statistics to the server. On the server, we sample initial noise containing the client's local distribution information from the fitted GMM, guide the DM to generate a synthetic dataset that complies with the client's distribution, and train the aggregated model.

In our method, the server has a pre-trained DM. The most widely used pre-trained DM is Stable Diffusion (Rombach et al., 2022), which consists of an encoder $\mathcal{E}$, a U-Net $\epsilon_\theta$, and a decoder $\mathcal{D}$. During the generation process, Stable Diffusion first maps the initial noise into the latent space via $\mathcal{E}$. Then, $\epsilon_\theta$ performs a denoising task over $T$ time steps, obtaining a noise-free latent representation, which is then decoded into a real image.

## 3.2 LOCAL DISTRIBUTION FITTING

The first step of our method is local distribution fitting. In this step, we need to fit the distribution of the noised client images and upload the statistics of the fitted distribution to the server. The sample noises have information of the local distributions, allowing the guidance on the server's DM to generate synthetic data that complies with the client data distribution. Specifically, the task at the client consists of two parts: Distribution Extraction and Distribution Fitting.

**Distribution Extraction.** We send the encoder $\mathcal{E}$ of the server's DM to the clients. The clients use $\mathcal{E}$ to encode their local images, obtaining the latent space representation of the image. Next, we add noise to these latent space representations to simulate the noise addition process in the DM, obtaining an initial noise distribution that reflects the client's image distribution. For client $m$, the client $m$ uses $\mathcal{E}$ to encode its local dataset $\mathbf{D}_m = \{(x_i)\}_{i=1}^{R}$, and obtains the latents of the local dataset:

$$\mathbf{z}_i = \mathcal{E}(x_i), x_i \in \mathbf{D}_m, i \in \{1, ..., R\} \tag{2}$$

Then, we simulate the forward process of the DM and add noise to these latents $z_i$, obtaining the noised local distributions. The noise addition process is as follows:

$$\bar{\mathbf{z}}_i = \sqrt{\bar{a}_T}\mathbf{z}_i + \sqrt{1 - \bar{a}_T}\epsilon, \tag{3}$$

where $T$ is the maximum time step of the DM, $\bar{a}_T$ is the hyperparameter related to the maximum time step $T$, which is determined by the DM, and $\epsilon$ is the random noise sampled from a standard Gaussian distribution $\mathcal{N}(0, \mathcal{I})$. Through this process, we obtain the $\bar{\mathbf{z}}_i$ from the noised local distribution of the clients. Due to privacy and communication concerns, we cannot directly upload these noised samples

to the server as the initial noise to guide the conditional generation of the DM. Therefore, we need to fit the distribution of these noised samples.

**Distribution Fitting.** We use a Gaussian Mixture Model (GMM) to fit the distribution of the noised client images for the following reasons: GMM represents the data distribution as a weighted sum of multiple Gaussian components, offering the advantage of being structurally simple and computationally inexpensive. At the same time, GMM has a relatively small number of parameters, which not only reduces computational and communication costs. Since the forward process is usually simulated by adding Gaussian noise, GMM, as a mixture of Gaussian distributions, can closely match the noised local distributions. Although it is possible to either train generative models on clients or upload noised client images (Chen et al., 2024), we argue that the former incurs excessive computational costs on the clients, while the latter poses greater risks of privacy leakage. Both approaches would negatively impact the practicality of the proposed method.

For the noised image samples $\bar{\mathbf{z}}_i, (i = 1, 2, \ldots, R)$, we first define the Gaussian Mixture Model as:

$$p_m(\bar{\mathbf{z}}|\pi_k^m, \mu_k^m, \Sigma_k^m) = \sum_{k=1}^{K} \pi_k^m \mathcal{N}(\bar{\mathbf{z}}|\mu_k^m, \Sigma_k^m) \tag{4}$$

where $\pi_k^m$ is the mixture weight of the $k$-th Gaussian component, $\mu_k^m$ and $\Sigma_k^m$ are the mean and covariance matrix of the component, and $\mathcal{N}(\bar{\mathbf{z}}|\mu_k^m, \Sigma_k^m)$ is the probability density function of the $k$-th Gaussian component.

To estimate the statistics of the GMM $(\pi_k^m, \mu_k^m, \Sigma_k^m)$, we use the Expectation-Maximization (EM) algorithm (Do & Batzoglou, 2008). In the E-step, we compute the posterior probability of each noised sample $\bar{\mathbf{z}}_i$ belonging to each Gaussian component:

$$\gamma_{ik} = \frac{\pi_k^m \mathcal{N}(\bar{\mathbf{z}}_i|\mu_k^m, \Sigma_k^m)}{\sum_{j=1}^{K} \pi_j^m \mathcal{N}(\bar{\mathbf{z}}_i|\mu_j^m, \Sigma_j^m)} \tag{5}$$

where $\gamma_{ik}$ represents the probability that the noised sample $\bar{\mathbf{z}}_i$ belongs to the $k$-th component. In the M-step, we update the parameters of the Gaussian Mixture Model:

$$\pi_k^m = \frac{1}{R} \sum_{i=1}^{R} \gamma_{ik}, \mu_k^m = \frac{\sum_{i=1}^{R} \gamma_{ik} \bar{\mathbf{z}}_i}{\sum_{i=1}^{R} \gamma_{ik}}, \Sigma_k^m = \frac{\sum_{i=1}^{R} \gamma_{ik}(\bar{\mathbf{z}}_i - \mu_k^m)(\bar{\mathbf{z}}_i - \mu_k^m)^T}{\sum_{i=1}^{R} \gamma_{ik}} \tag{6}$$

These update steps adjust the model's parameters based on the current responsibility values $\gamma_{ik}$ to maximize the log-likelihood function of the data. The E-step and M-step are repeated until the model's parameters converge. The client will then upload the Gaussian component parameters ($\mu_k^m$, $\Sigma_k^m$, $\pi_k^m$) containing the client distribution information to the server. This information can be used to generate initial noise that matches the client distribution, enabling the server's DM to generate a synthetic dataset that matches the client distribution.

Through the above process, the GMMs fitted locally at the clients can provide initial noises containing information of the noised local distributions, which in turn helps the server generate a synthetic dataset that complies with the local distribution, enabling the trained aggregated model to generalize across the different client distributions.

## 3.3 SERVER IMAGE GENERATION

Once the server receives the fitted GMMs, we sample initial noises that comply with the client distributions from these GMMs. This initial noise serves as the starting point of the diffusion process to guide the server's DM to generate data that complies with the client distributions.

Firstly, for client $m$, the server needs to randomly sample from the Gaussian Mixture Model $p_m(\bar{\mathbf{z}}|\pi_k^m, \mu_k^m, \Sigma_k^m)$ to generate the initial noise $\mathbf{x}_T$. The distribution of the sampled initial noised is consistent with the noised local distribution of the clients. Specifically, the sampling process from the GMM is as follows:

$$\mathbf{x}_T \sim \sum_{k=1}^{K} \pi_k^m \mathcal{N}(\mathbf{x}_T|\mu_k^m, \Sigma_k^m) \tag{7}$$

Table 1: The performances of the compared methods on OpenImage, DomainNet, and NICO++ under the non-IID feature distribution skew, where the italicized texts represent the performance ceiling of centralized training used as a reference, and bold texts represent the best performance of the compared methods.

| | OpenImage | | | | | | | DomainNet | | | | | | |
|---|---|---|---|---|---|---|---|---|---|---|---|---|---|---|
| | client0 | client1 | client2 | client3 | client4 | client5 | average | clipart | infograph | painting | quickdraw | real | sketch | average |
| *Ceiling* | *49.88* | *50.56* | *57.89* | *59.96* | *66.53* | *51.38* | *56.03* | *47.48* | *19.64* | *45.24* | *12.31* | *59.79* | *42.35* | *36.89* |
| FedAvg | 42.48 | 47.24 | 47.01 | 51.28 | 61.87 | 45.47 | 49.22 | 37.96 | 12.55 | 34.41 | 5.93 | 51.33 | 32.37 | 29.09 |
| FedDF | 43.26 | 44.98 | 52.54 | 56.71 | 62.89 | 48.37 | 51.45 | 38.09 | 13.68 | 35.48 | 7.32 | 53.83 | 34.69 | 30.51 |
| FedProx | 44.99 | 48.83 | 49.25 | 56.68 | 61.23 | 46.07 | 51.17 | 38.24 | 12.46 | 37.29 | 6.26 | 54.88 | 35.76 | 30.81 |
| FedDyn | 46.93 | 46.08 | 52.44 | 54.67 | 62.84 | 47.73 | 51.78 | 40.12 | 14.77 | 36.59 | 7.73 | 54.85 | 34.81 | 31.47 |
| Prompts Only | 32.91 | 33.24 | 41.72 | 45.02 | 49.85 | 35.97 | 39.78 | 31.80 | 11.61 | 31.14 | 4.13 | 61.53 | 31.44 | 28.60 |
| FedDISC | 47.42 | 49.65 | 54.73 | 53.41 | 60.74 | 52.81 | 53.12 | 43.89 | 14.84 | 38.38 | 8.35 | 56.19 | 36.82 | 33.07 |
| FGL | 48.21 | 49.16 | 54.98 | 55.47 | 63.14 | 49.32 | 53.38 | 41.81 | 15.30 | 40.67 | 8.79 | 57.58 | 39.54 | 33.94 |
| FedLMG | 48.99 | 51.66 | 55.59 | 52.80 | 62.41 | 58.86 | 55.05 | 44.25 | 17.51 | 38.74 | 9.43 | 57.31 | 38.44 | 34.28 |
| FedDEO | 51.08 | **52.53** | 61.22 | **62.18** | **67.31** | 56.68 | 58.50 | 46.77 | 18.28 | 43.97 | 10.73 | 60.64 | 41.45 | 36.08 |
| FedLD | **52.25** | 51.55 | **63.08** | 60.77 | 66.43 | **58.49** | **58.76** | **48.63** | **21.74** | **45.33** | **15.04** | **62.51** | **45.65** | **38.65** |

| | Common NICO++ | | | | | | | Unique NICO++ | | | | | | |
|---|---|---|---|---|---|---|---|---|---|---|---|---|---|---|
| | autumn | dim | grass | outdoor | rock | water | average | client0 | client1 | client2 | client3 | client4 | client5 | average |
| *Ceiling* | *62.66* | *54.07* | *64.89* | *63.04* | *61.08* | *54.63* | *60.06* | *79.16* | *81.51* | *76.04* | *72.91* | *79.16* | *79.29* | *78.01* |
| FedAvg | 52.51 | 40.45 | 57.21 | 51.59 | 49.31 | 43.56 | 49.11 | 67.31 | 74.73 | 69.01 | 64.37 | 73.07 | 67.87 | 69.39 |
| FedDF | 50.44 | 39.62 | 57.42 | 52.91 | 51.61 | 44.76 | 49.46 | 69.79 | 78.90 | 69.53 | 66.01 | 74.86 | 70.80 | 71.64 |
| FedProx | 53.49 | 42.41 | 58.84 | 53.08 | 53.67 | 45.42 | 51.15 | 70.46 | 75.30 | 70.87 | 67.67 | 72.84 | 71.51 | 71.44 |
| FedDyn | 54.38 | 43.20 | 57.56 | 52.63 | 52.86 | 46.76 | 51.23 | 71.23 | 74.98 | 69.68 | 68.13 | 73.63 | 70.61 | 71.37 |
| Prompts Only | 50.49 | 38.10 | 54.53 | 49.39 | 49.12 | 41.58 | 47.20 | 69.79 | 69.14 | 69.32 | 59.89 | 67.83 | 66.42 | 67.06 |
| FedDISC | 56.82 | 51.43 | 59.45 | 56.17 | 52.32 | 45.64 | 53.64 | 74.32 | 73.47 | 71.25 | 66.79 | 75.28 | 70.06 | 71.86 |
| FGL | 57.25 | 49.35 | 61.81 | 58.42 | 54.29 | 47.62 | 54.79 | 74.62 | 79.43 | 71.26 | 68.65 | 76.37 | 74.31 | 74.10 |
| FedLMG | 54.63 | 49.21 | 58.13 | 54.75 | 54.64 | 47.03 | 53.06 | 75.13 | 73.30 | 70.31 | 68.88 | 73.60 | 72.51 | 72.28 |
| FedDEO | 71.03 | **58.02** | 73.33 | **68.53** | 68.16 | 63.04 | 67.01 | 81.25 | 86.19 | 82.94 | 79.94 | 83.85 | 80.27 | 82.40 |
| FedLD | **73.35** | 57.76 | **76.17** | 68.48 | **70.24** | **66.26** | **68.71** | **82.59** | **87.06** | **85.35** | **81.08** | **85.63** | **82.84** | **84.09** |

Based on the initial noise $\mathbf{x}_T$ sampled from the GMM and the category text prompt $p_c$ that ensures the correct semantic information, the server's DM $\epsilon_\theta$ gradually transforms the noise into the final image $\mathbf{x}_0$ through multiple iterative steps. The generation process at each time step $t \sim \{1, ..., T\}$ can be expressed as:

$$\mathbf{x}_{t-1} = \sqrt{\alpha_{t-1}}\Big(\frac{\mathbf{x}_t - \sqrt{1-\alpha_t}\epsilon_\theta(\mathbf{x}_t|p_c)}{\sqrt{\alpha_t}}\Big) + \sqrt{1-\alpha_{t-1}-\sigma_t^2}\epsilon_\theta(\mathbf{x}_t|p_c) + \sigma_t\boldsymbol{\varepsilon}_t \tag{8}$$

where $\alpha_t$ and $\sigma_t$ are the denoising coefficients predefined by the DM, and $\boldsymbol{\varepsilon}_t$ is the random noise sampled from $\mathcal{N}(0, \mathcal{I})$ at each time step $t$. As the denoising process progresses, the noise is gradually transformed into a semantically coherent image $\mathbf{x}_0$, which is then decoded into a real image by the server's decoder $\mathcal{D}$.

Through this process, the server can generate a synthetic dataset $\mathbf{D}_{\text{gen}}$ containing a large number of image samples that comply with the client's distribution. The category prompt $p_c$ used in the generation process naturally provides pseudo-labels for these synthetic images, which can then be used with the cross-entropy loss function $\mathcal{L}_{CE}$ to train the aggregated model $\mathcal{F}_g$. Since the initial noise in the DM is sampled from the GMM, which fits the noisy local image distribution of the client well, this noise inherently contains information about the client's personalized distribution. This enables the generated synthetic dataset to effectively reflect the client's localized personalized features, thereby allowing the aggregated model to adapt to the data distribution of each client. The subsequent experimental section will provide a quantitative analysis and visual demonstration of the performance of the synthetic dataset and the aggregated model.

# 4 EXPERIMENTS

In this section, we conduct comparisons with a large number of SOTA methods under detailed experimental settings. We consider various types of differences that clients may exhibit. We also conduct ablation experiments to further demonstrate the impact of each module and hyperparameter on the performance. Furthermore, we delve into several issues that may affect its practical implementation, including privacy, communication costs, and computational costs.

## 4.1 EXPERIMENTAL SETTING

**Datasets and Implementation Details.** This study conducts experiments on three datasets: **Domain-Net**(Peng et al., 2019), **OpenImage**(Kuznetsova et al., 2020), and **NICO++**(Zhang et al., 2022b).

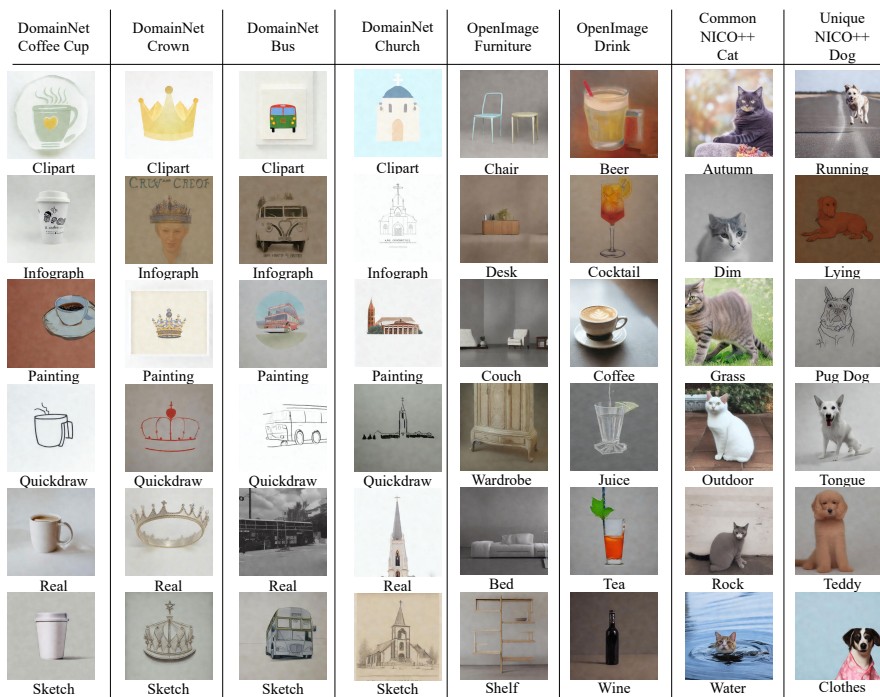

Figure 2: Visualization of the FedLD synthetic dataset. FedLD can effectively address the style differences across different clients.

**DomainNet** is used to simulate style variations within categories, **OpenImage** is used to simulate subclass differences within supercategories, and **NICO++** is used to simulate changes in background and specific object attributes. It is important to emphasize that although each domain in these datasets is accompanied by textual descriptions, we use only the category labels as text prompts in all generation processes to enhance practicality. In practical use, clients cannot be expected to provide textual descriptions of their personalized data distributions. Detailed information about the datasets and implementation is provided in Appendix B.

**Client Partition.** The client partition design aims to simulate data heterogeneity among clients, especially in non-IID scenarios. In federated learning, data heterogeneity can mainly be divided into two types: feature heterogeneity and label heterogeneity (Kairouz et al., 2021). To this end, we address both types of heterogeneity in client partitioning. In the feature heterogeneity scenario, we conduct experiments on all four datasets. For each dataset, the six data domains for all categories are assigned to six clients, with each client only possessing a specific data domain for all categories. In the case of label heterogeneity, experiments are conducted on the Common NICO++ and Unique NICO++ datasets. In this experimental setup, we divide the 60 categories into six groups of 10 categories each, which are distributed across the six clients, with each client containing data from all 10 categories. As stated in this paper, there is no overlap of data between clients in all partition schemes. Therefore, the adopted partition method greatly enhances the data heterogeneity among clients while covering various client data heterogeneity scenarios.

**Comparison Methods.** We compare our method with nine other methods, which can be divided into three main categories: 1. **Performance Upper Bound (Ceiling).** The performance upper bound for traditional federated learning methods is centralized training, i.e., uploading all clients' local data for training the aggregation model. 2. Traditional federated learning methods with multiple communication rounds: **FedAvg (McMahan et al., 2017), FedDF (Lin et al., 2020), FedProx (Li et al., 2020), FedDyn (Acar et al., 2021)**. These methods use 20 communication rounds, where each round involves training by each client. In FedDF, we used ImageNet as additional public data for distillation. 3. Diffusion-based OSFL methods: **FedDISC (Yang et al., 2023)**, **FedLMG (Yang et al., 2025)**, **FedDEO (Yang et al., 2024)**, **FGL (Zhang et al., 2023)**, and **Prompts Only**. Although FedDISC was designed for semi-supervised FL scenarios, we removed the pseudo-labeling process in

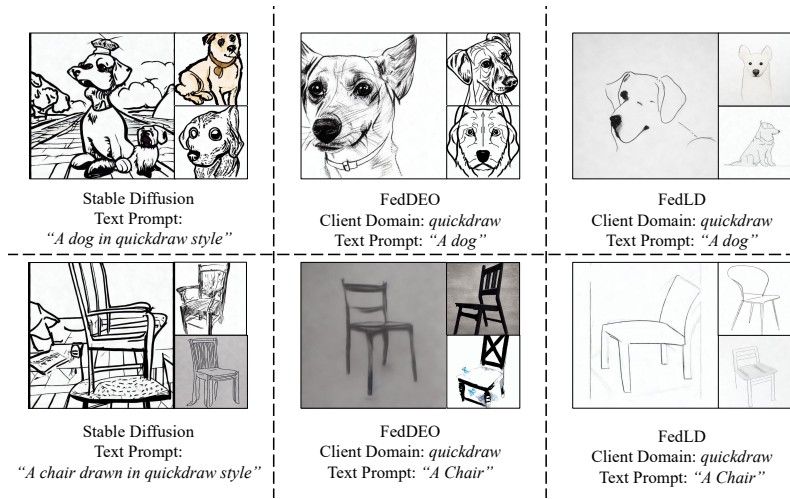

Figure 3: Visualization comparison of FedLD on the *quickdraw* domain.

Table 2: Comparison of the communication cost.

| Upload Communication Cost (M) | | | | | |
|---|---|---|---|---|---|
| FedAvg | Ceiling | FedLMG | FedDISC | FedDEO | FedLD |
| $20 \times 11.69 = 233.8$ | 270.95 | 11.69 | 4.23 | 3.54 | **1.31** |
| download Communication Cost (GB) | | | | | |
| FedAvg | CLIP-based FL | FedDISC | FGL | FedDEO | FedLD |
| 2.28 | 1.71 | 1.74 | 1.88 | 1.72 | **0.16** |

FedDISC and directly used the true labels of client images. Another point to note is **Prompts Only**, where the server completely relies on textual prompts from the category names to generate images, without using any local client description data, representing a generation strategy that is purely based on textual prompts without client information guidance. It is worth mentioning that some other OSFL methods, such as DENSE (Zhang et al., 2022a) and FedCVAE (Heinbaugh et al., 2022), are not used for comparison due to the difficulty of training the generator to convergence in high-resolution real image scenarios.

## 4.2 MAIN RESULTS

**Quantitative Experiments.** Table 1 shows the performance comparison of FedLD with other methods on three datasets under feature distribution skew. We highlight several observations: 1) FedLD performs best on various clients. In most cases, it consistently outperforms the performance ceiling of centralized training (*Ceiling*). The main reason lies in the fact that DMs introduce additional knowledge into the OSFL framework, which helps address common issues such as class imbalance and insufficient data quality, whereas our method can effectively improve model performance by synthesizing balanced and high-quality data on the server. 2) In domains with unique backgrounds and styles, such as **DomainNet**'s *quickdraw* and *sketch*, FedLD shows a substantial performance improvement, surpassing all compared methods, demonstrating that compared to other attention-based guidance methods, the guidance of initial noise can more effectively control the background and outlines of the generated object. 3) On some domains of the OpenImage dataset, FedLD's performance is slightly lower than that of FedDEO, due to the subclass differences in the OpenImage dataset, which manifest primarily in the detailed semantics of the images. In this case, attention-based guidance methods like FedDEO tend to capture these fine-grained differences more effectively. 4) In comparison with the Prompts Only method, FedLD shows a significant advantage, demonstrating that fitting the personalized distribution of clients significantly improves the performance of the aggregation model. Without any guidance, the synthetic dataset tends to converge towards certain common domains, such as *real* in **DomainNet**.

Table 3: Comparison of the computation cost.

| Client Computation Cost (GFLOPS) | | | | |
|---|---|---|---|---|
| CLIP-based Federated Fine-tuning | FedDISC | FGL | FedDEO | FedLD |
| 493.5 | 334.73 | 227.34 | 365.72 | **208.55** |

**Visualization Experiments.** We visualize the synthetic images guided by different client distributions, where each client corresponds to a distinct style. As shown in Figures 2, we can observe that our method generates high-quality realistic images, with great adaptability of the heterogeneity in styles, backgrounds, subclasses, and detailed semantics. In comparison with FedDEO and Stable Diffusion with specific text prompt in Figure 3, FedLD outperforms both, offering better control over background and lines, and producing more detailed, client-tailored images. This highlights FedLD's superior ability to generate high-quality, personalized images.

To further validate the performance of our method, we conduct extensive ablation experiments to discuss in detail the impact of hyperparameters and other settings on our method, including the number of images in the synthetic dataset, the number of training epochs for local descriptions, the pre-trained DM used, and the number of clients. Due to the space limitation, most of the ablation experiments are provided in the supplementary material C.

## 4.3 OTHER DISCUSSIONS

**Communication and Computation Costs.** Communication cost, including upload and download, is a key metric in federated learning. We compare upload model parameters and the total size of downloaded files for different methods. FedAvg, FedDF, FedProx, and FedDyn have similar costs; the Prompts Only method is excluded due to no communication cost; FGL is excluded due to random prompt length. So they are not detailed. As shown in Table 2, FedLD outperforms other methods in both upload and download costs. FedLD only requires the image encoder and uploads fewer parameters, reducing communication. These results highlight FedLD's efficiency, especially in bandwidth-limited large-scale federated learning scenarios.

We also compare the computation costs of the compared methods. Regarding server computation cost, FedLD and other diffusion-based OSFL methods have similar server costs. In federated learning, the server is typically assumed to have sufficient computational power, while client devices are usually limited (Li et al., 2020). Therefore, we focus on comparing the client's computation costs. In table 3, it can be seen that since FedLD only needs to fit the Gaussian Mixture Model on the clients and does not require complex model training or large-scale parameter optimization, the computational cost on the client is significantly lower than that of other methods.

**Privacy Protection.** FedLD provides strong privacy protection. It uploads only the statistics of the fitted image distribution, not specific image features, sufficiently reducing privacy risks. Additionally, FedLD works with noised image distributions, adding noise to enhance privacy and prevent reverse engineering. The minimal upload of GMM parameters reduces both data overhead and privacy risks. Although methods guided by initial noise still carry a minimal risk of directly reconstructing the client's original images, recent works including (Chen et al., 2024) explicitly upload the noised images from clients, which clearly involves a greater degree of privacy leakage compared to our method. We provide further discussion on privacy issues in the supplementary material C.3.

## 5 CONCLUSION

This paper introduces FedLD, a local distribution conditioned diffusion for OSFL. FedLD uses the Gaussian Mixture Model to fit the clients' noised image distributions and guide a DM in generating synthetic datasets, reducing communication and computation costs while avoiding the risk of privacy. FedLD outperforms all compared methods, including centralized training, and handling heterogeneous clients by creating high-quality, personalized datasets. It offers an efficient, privacy-preserving solution for federated learning, ideal for resource-constrained, large-scale devices, and further showcases the potential of DMs in OSFL.

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

## A  LLMs Usage Statement

In preparing this manuscript, large language models (LLMs) were used exclusively for linguistic refinement. Specifically, we employed OpenAI's GPT-5 model to polish the wording and improve the readability of the text. No part of the conceptualization, methodology, experimental design, analysis, or conclusions was generated by or relied upon LLMs. All scientific contributions are the original work of the authors.

## B  Experimental Setting Details

We detail the experimental settings of the proposed method that couldn't be fully elaborated in the main text due to space constraints. These primarily include: (1) Dataset Details, (2) The Number of Images, and (3) Implementation Details.

### B.1  Dataset Details

The experiments were carried out on three datasets: **DomainNet**(Peng et al., 2019), **OpenImage**(Kuznetsova et al., 2020), and **NICO++**(Zhang et al., 2022b). All datasets contain large-scale real-world images with a resolution of 224x224 pixels. **DomainNet** consists of six domains: *clipart*, *infograph*, *painting*, *quickdraw*, *real*, and *sketch*, with each domain containing 345 categories. For the **OpenImage** dataset, we followed the partition used in FedDISC(Yang et al., 2023), which divides it into 20 supercategories based on the hierarchical structure of categories provided by OpenImage. Each supercategory includes 6 fine-grained subclasses, which serve as the data domains for each category. A detailed overview of the OpenImage partition can be found in Table 5. **NICO++** contains 60 categories, each with six common domains shared across categories, as well as six unique domains specific to each category. These two configurations are referred to as the **Unique NICO++** and **Common NICO++** datasets. For example, in the Common NICO++ dataset, both the *Cat* and *Dog* categories contain six data domains: *autumn*, *dim*, *grass*, *outdoor*, *rock*, and *water*. In the Unique NICO++ dataset, the *Cat* class includes six unique domains: *Eating*, *In Cave*, *In Mud*, *Jumping*, *Maine Cat*, and *Walking*, while the *Dog* class contains six distinct data domains: *Lying*, *Pug Dog*, *Running*, *Sticking Out Tongue*, *Teddy Dog*, and *Wearing Clothes*.

Example images from each dataset are shown in Figure 4. As highlighted in the main text, the figure effectively demonstrates the varying emphasis placed on the partitioning of data domains across the different datasets. **DomainNet** mainly emphasizes image style, while **OpenImage** focuses on fine-grained subcategories within each supercategory. **Common NICO++** emphasizes image backgrounds, and **Unique NICO++** highlights specific object attributes. These datasets collectively capture a wide range of potential differences that may arise among clients, thereby enhancing the applicability and relevance of the proposed method.

### B.2  The Number of Images

In our experimental setup, the number of images in each client's dataset plays a crucial role, as the scale of the client's data directly impacts the training effect and the performance of the aggregation model. To evaluate the effectiveness of the proposed method, we compare it with **Ceiling** (i.e., the centralized training performance upper bound), which represents the maximum possible performance of the model when all data is centralized for training in a centralized training scenario. To achieve this comparison, we ensure a fair comparison between the synthetic dataset and the original client dataset, so we need to ensure that the number of images for each client in the experiment is representative, while the server generates the same number of images as the number of clients. With this setup, we can guarantee that regardless of the size of the client's local dataset, the synthetic dataset will match the original client dataset in both quantity and category distribution, thereby eliminating any potential impact of data imbalance on the experimental results. This approach makes the experiment fairer and better reflects the performance of the proposed method under different client data conditions. Considering the generation cost and practical feasibility of the experiment, we set the number of images generated for each description to 30 in most experiments. This setting is closer to practical scenarios because many clients in real-world environments may not have enough labeled data, especially those with data scarcity. Therefore, using 30 images as the size of each client's

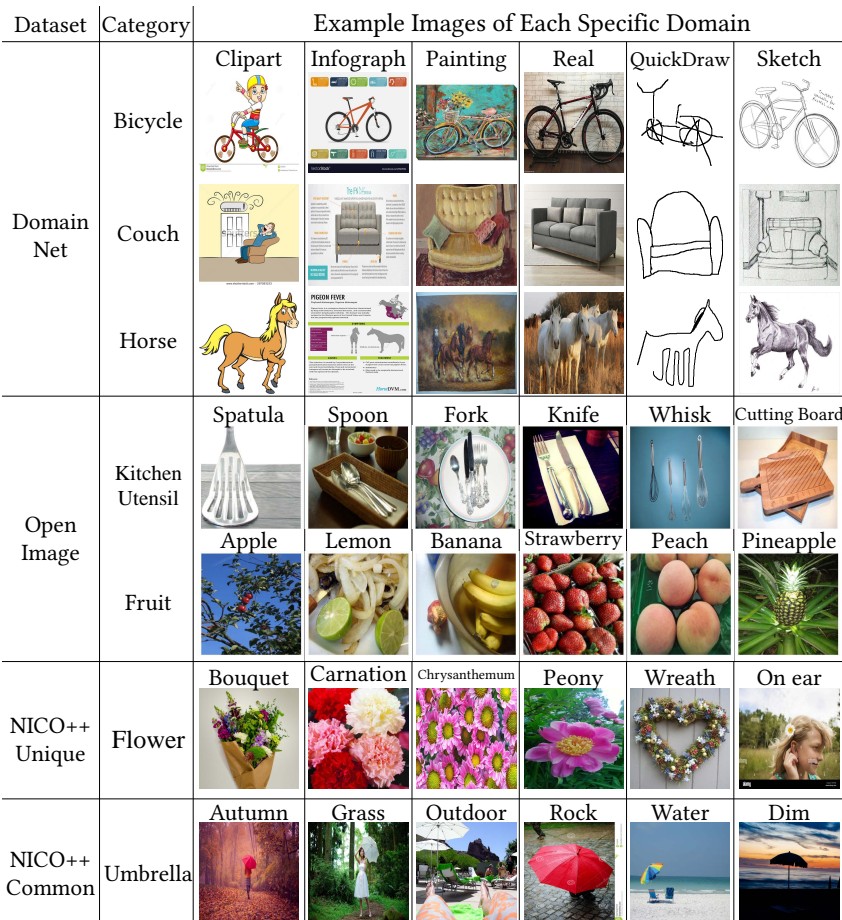

Figure 4: The example images of used datasets.

synthetic dataset ensures the effectiveness of the synthetic dataset while keeping the generation cost manageable, avoiding excessive computational overhead.

## B.3 IMPLEMENTATION DETAILS

In our experiments, we mainly use ResNet-18 (He et al., 2016) as the structure for the aggregation model. The primary pre-trained DM used is *Stable-diffusion-v1.5*, from the *HuggingFace* model library, which includes a corresponding CLIP text encoder for extracting the text features $f_c$ for each category $c$. We also use *Stable-diffusion-v2.1* from the *HuggingFace* model library, with the pre-trained *Latent DM*(Rombach et al., 2022), from *Github*. Both *Stable-diffusion-v1.5* and *Stable-diffusion-v2.1* were pre-trained on the LAION-5B dataset(Schuhmann et al., 2022), and the *Latent DM* was pre-trained on the LAION-400M dataset (Schuhmann et al., 2021). These two datasets are large-scale image-text paired datasets covering a wide range of image distributions encountered in everyday life. All experiments were conducted on four NVIDIA GeForce RTX 3090 GPUs. Regarding the specific hyperparameter settings, the number of images generated for each description $R$ is set to 30, unless specified otherwise in ablation experiments related to the number of images. Relevant hyperparameters in the diffusion generation process are set to their default values. The inference steps are 50, and the generated guiding coefficient is 3.

Table 4: Results on the impact of the number of generated images on FedLD. $R$ represents the number of generated images, with bold text indicating the best performance.

| | DomainNet | | | | | | |
|---|---|---|---|---|---|---|---|
| | *clipart* | *infograph* | *painting* | *quickdraw* | *real* | *sketch* | average |
| $R$=10 | 45.64 | 18.71 | 44.01 | 12.44 | 61.27 | 44.72 | 36.41 |
| $R$=30 | 48.63 | 21.74 | 45.33 | 15.04 | 62.51 | 45.64 | 38.65 |
| $R$=50 | **49.43** | **23.98** | **47.88** | **18.64** | **63.27** | **46.02** | **41.57** |

Table 5: The client partition on the OpenImage dataset.

| Supercategory | Baked Goods | Bird | Building | Carnivore | Clothing | Drink | Fruit | Furniture | Home appliance | Human body |
|---|---|---|---|---|---|---|---|---|---|---|
| Client0 | Pretzel | Woodpecker | Convenience Store | Bear | Shorts | Beer | Apple | Chair | Washing Machine | Human Eye |
| Client1 | Bagel | Parrot | House | Leopard | Dress | Cocktail | Lemon | Desk | Toaster | Skull |
| Client2 | Muffin | Magpie | Tower | Fox | Swimwear | Coffee | Banana | Couch | Oven | Human Mouth |
| Client3 | Cookie | Eagle | Office Building | Tiger | Brassiere | Juice | Strawberry | Wardrobe | Blender | Human Ear |
| Client4 | Bread | Falcon | Castle | Lion | Tiara | Tea | Peach | Bed | Gas Stove | Human Nose |
| Client5 | Croissant | Sparrow | Skyscraper | Otter | Shirt | Wine | Pineapple | Shelf | Mechanical Fan | Human Foot |
| Supercategory | Kitchen Utensil | Land Vehicle | Musical Instrument | Office Supplies | Plant | Reptile | Sports Equipment (Ball) | Toy | Vegetable | Weapon |
| Client0 | Spatula | Ambulance | Drum | Pen | Maple | Dinosaur | Football | Doll | Potato | Knife |
| Client1 | Spoon | Cart | Guitar | Poster | Willow | Lizard | Tennis Ball | Balloon | Carrot | Axe |
| Client2 | Fork | Bus | Harp | Calculator | Rose | Snake | Baseball | Dice | Broccoli | Sword |
| Client3 | Knife | Van | Piano | Whiteboard | Lily | Tortoise | Golf Ball | Flying Disc | Cabbage | Handgun |
| Client4 | Whisk | Truck | Violin | Box | Common Sunflower | Crocodile | Rugby Ball | Kite | Bell Pepper | Shotgun |
| Client5 | Cutting Board | Car | Accordion | Envelope | Houseplant | Sea Turtle | Volleyball | Teddy Bear | Pumpkin | Dagger |

# C SUPPLEMENTARY EXPERIMENTS

## C.1 EXPERIMENTS UNDER LABEL DISTRIBUTION SKEW

We also conduct experiments under Label Distribution Skew to validate the performance of FedLD. As shown in Table 6, FedLD is applicable in scenarios with client label heterogeneity, surpassing all comparison methods and the centralized training ceiling on most clients, further demonstrating its adaptability to various client data heterogeneity issues.

## C.2 ABLATION EXPERIMENTS

### C.2.1 NUMBER OF IMAGES

The number of images in the synthetic dataset is a key factor influencing the performance of our method. To explore the impact of $R$ on the method's performance, we conduct ablation experiments on the DomainNet dataset, with the results shown in Table 4. We can see that as $R$ increases, the performance of the trained aggregation model improves significantly. This indicates that the generated synthetic dataset has been effectively enhanced in terms of diversity and coverage, promoting the generalization and stability of the model. Notably, when $R$ increased from 30 to 50, the rate of performance improvement did not significantly slow down, demonstrating that the diversity of the synthetic dataset continuously drives performance. In other words, as the number of images increases, the synthetic dataset, while covering more samples and transformations, still brings gradual performance improvements, suggesting that generating more images benefits model training without hitting a performance plateau.

### C.2.2 THE USED DIFFUSION MODEL

The DM used to generate the synthetic dataset is a key component of our method. However, this does not imply that our method heavily depends on a specific DM. To verify this, we conducted ablation experiments in the feature heterogeneity scenario on three commonly used DMs: *Stable-diffusion-v1.5*, *Stable-diffusion-v2.1*, and *Latent DM* (LDM). The experimental results are shown in Table 7. The following important conclusions can be drawn from the table: (1) Different DMs can all train aggregation models with high performance using our method, confirming the flexibility of the method in selecting different DMs. Regardless of the DM used, the generated synthetic dataset effectively supports the training of the aggregation model and achieves good performance. (2) Although the aggregation model trained with LDM outperforms the Ceiling (performance upper bound), its overall performance is poor. This gap is mainly due to LDM's relatively small pretraining scale and parameter count, which leads to a low overlap with client distributions. Therefore, while LDM can generate some synthetic data, its performance in terms of image quality and diversity

Table 6: The performances of the compared methods on OpenImage, DomainNet, and NICO++ under the non-IID label distribution skew, where the italicized texts represent the performance ceiling of centralized training used as a reference, and bold texts represent the best performance of the compared methods.

| | Common NICO++ | | | | | | Unique NICO++ | | | | | |
|---|---|---|---|---|---|---|---|---|---|---|---|---|
| | client0 | client1 | client2 | client3 | client4 | client5 | average | client0 | client1 | client2 | client3 | client4 | client5 | average |
| *Ceiling* | *50.24* | *54.36* | *63.35* | *64.82* | *61.99* | *65.09* | *59.98* | *74.02* | *78.9* | *79.68* | *74.47* | *77.34* | *77.47* | *76.98* |
| FedAvg | 18.23 | 27.79 | 36.32 | 52.42 | 37.96 | 39.24 | 35.32 | 34.96 | 58.98 | 38.41 | 63.41 | 45.44 | 59.76 | 50.16 |
| FedDF | 31.40 | 32.22 | 43.73 | 45.19 | 36.01 | 43.08 | 38.60 | 51.85 | 52.34 | 55.85 | 52.47 | 54.42 | 59.24 | 54.36 |
| FedProx | 37.31 | 35.95 | 42.78 | 48.92 | 41.07 | 47.53 | 42.26 | 54.55 | 60.51 | 54.05 | 58.34 | 55.69 | 57.78 | 56.82 |
| FedDyn | 36.83 | 37.85 | 45.21 | 51.38 | 42.74 | 44.36 | 43.06 | 55.29 | 59.71 | 56.68 | 61.74 | 48.99 | 61.31 | 57.28 |
| Prompts Only | 38.64 | 45.55 | 53.08 | 54.72 | 50.19 | 59.91 | 50.34 | 67.38 | 71.88 | 67.70 | 64.19 | 63.41 | 63.28 | 66.30 |
| FedDISC | 50.75 | 51.64 | 60.79 | 58.33 | 55.41 | 57.28 | 55.70 | 71.89 | 73.20 | 70.51 | 70.02 | 75.62 | 69.82 | 71.84 |
| FGL | 45.34 | 51.41 | 60.44 | 59.65 | 58.87 | 62.33 | 56.34 | 69.51 | 74.59 | 71.36 | 69.41 | 69.65 | 71.42 | 70.99 |
| FedLMG | **58.98** | 46.53 | 60.93 | 57.45 | 53.92 | 54.32 | 55.35 | 73.30 | 71.48 | 68.97 | 69.71 | 72.91 | 65.49 | 70.31 |
| FedDEO | 53.69 | **56.46** | 66.32 | 66.57 | 62.26 | **70.81** | 62.68 | 76.58 | 80.42 | **81.19** | 75.75 | 80.38 | 78.94 | 78.87 |
| FedLD | 56.91 | 56.25 | **68.55** | **68.26** | **63.76** | 69.72 | **63.90** | **78.64** | **82.79** | 80.09 | **79.06** | **82.55** | 79.14 | **80.37** |

Table 7: Results on the impact of the used diffusion models on FedLD, with bold text indicating the best performance.

| | Common NICO++ | | | | | | |
|---|---|---|---|---|---|---|---|
| | autumn | dim | grass | outdoor | rock | water | average |
| *LDM* | 70.33 | 55.52 | 72.21 | 65.78 | 65.59 | 63.67 | 65.51 |
| *SD-v1.5* | 73.35 | 57.76 | 76.17 | 68.48 | 70.24 | 66.26 | 68.71 |
| *SD-v2.1* | **75.67** | **58.61** | **77.04** | **69.58** | 69.38 | **68.02** | **69.72** |
| | Unique NICO++ | | | | | | |
| | client 0 | client 1 | client 2 | client 3 | client 4 | client 5 | average |
| *LDM* | 73.47 | 80.71 | 79.35 | 75.28 | 80.19 | 78.22 | 77.87 |
| *SD-v1.5* | 82.59 | 87.06 | 85.35 | **81.08** | **85.63** | 82.84 | 84.09 |
| *SD-v2.1* | **83.88** | **88.45** | **86.74** | 80.08 | 84.16 | **83.21** | **84.42** |

does not match that of larger pre-trained models. (3) The best performance is achieved with *Stable-diffusion-v2.1*. Although both *Stable-diffusion-v1.5* and *Stable-diffusion-v2.1* were pre-trained on the LAION-5B dataset, *Stable-diffusion-v2.1* has more parameters and better generation quality, allowing it to more accurately capture client data distributions and generate higher-quality synthetic datasets, thereby improving the performance of the aggregation model. In summary, the experimental results show that FedLD is not dependent on a specific DM. While different DMs have some impact on the quality of the generated dataset and the performance of the aggregation model, FedLD performs well on various commonly used DMs, demonstrating its broad adaptability and flexibility. Therefore, FedLD can be widely used in different practical application scenarios, with strong adaptability and scalability.

### C.2.3 NUMBER OF CLIENTS

Based on the experimental setup outlined above, we perform ablation experiments to analyze the impact of the number of clients on the Common NICO++ and Unique NICO++ datasets in the label heterogeneity scenario. The number of clients is varied at 6, 30, 60, and 180. The results of these experiments are summarized in Table 8. It is worth noting that, for consistency with the previous experiments, we still use the naming convention "client 0–6" for the test domain subsets of Unique NICO++, indicating that the test sets are the same as those used in earlier experiments. The first column of the table, "The Number of Clients," denotes the variation in the number of clients. As shown in the table, the quality of the synthetic dataset and the performance of the trained aggregated model remain largely unaffected by increasing the number of clients, provided that the client models are well-trained. As emphasized in the main text, the proposed method demonstrates a high degree of independence across clients, exhibiting robust adaptability even as the number of clients grows. These results further highlight the method's practicality in large-scale client scenarios.

Table 8: Results of the ablation experiments on the number of clients.

| The Number of Clients | Common NICO++ | | | | | | | Unique NICO++ | | | | | | |
|---|---|---|---|---|---|---|---|---|---|---|---|---|---|---|
| | autumn | dim | grass | outdoor | rock | water | Avg | client0 | client1 | client2 | client3 | client4 | client5 | Avg |
| 6 | 56.91 | 56.25 | 68.55 | **68.26** | **63.76** | 69.72 | 63.91 | 78.64 | 82.79 | 80.09 | 79.06 | 82.55 | 79.14 | 80.37 |
| 30 | 55.46 | 56.93 | 68.87 | 67.03 | 62.88 | 68.74 | **64.89** | 78.76 | **83.68** | 80.55 | 80.79 | 83.32 | 79.04 | 81.02 |
| 60 | 56.58 | 57.31 | 68.67 | 66.97 | 62.22 | 70.23 | 63.66 | **79.57** | 83.43 | 79.35 | **81.44** | 82.24 | 78.05 | 80.72 |
| 180 | **57.75** | **57.89** | **69.35** | 67.44 | 63.24 | **69.86** | 64.25 | 79.43 | 82.19 | **81.46** | 80.08 | **84.74** | **80.09** | **81.33** |

Client Dataset             Synthetic Dataset

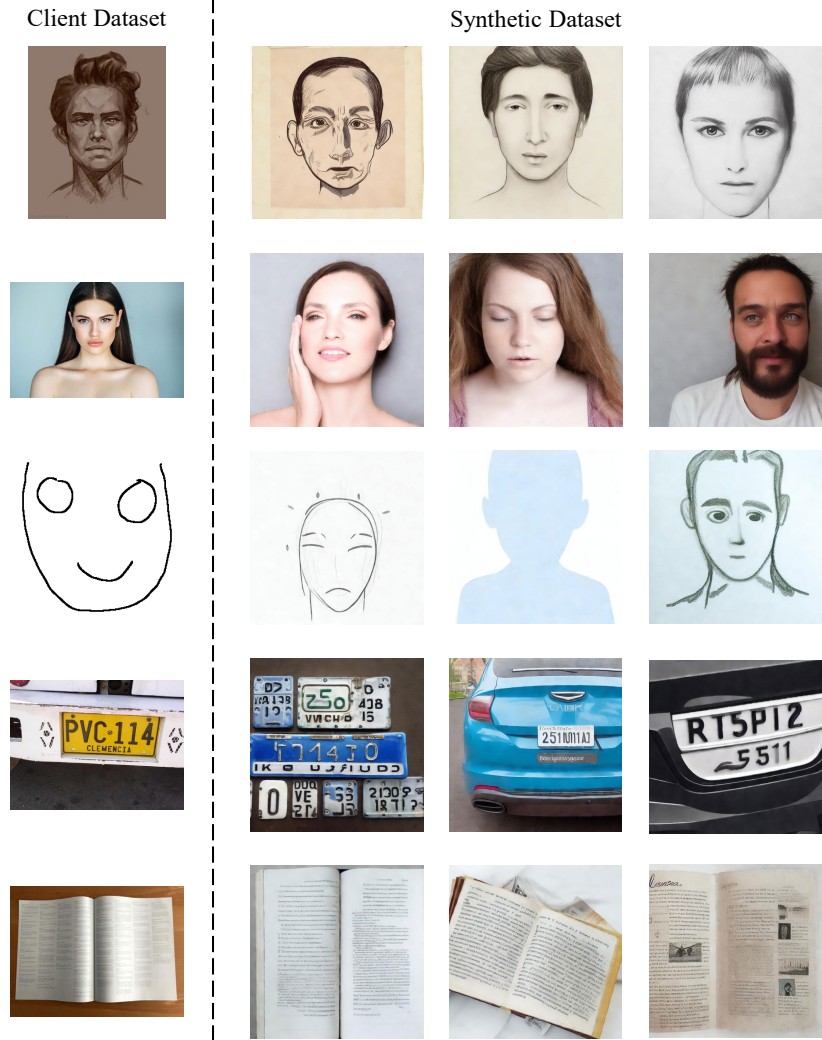

Figure 5: Visualization pf privacy-sensitive information- related categories.

## C.3 PRIVACY-RELATED DISCUSSION

To further demonstrate the privacy-protecting capabilities of our method, we conduct a series of visualization experiments. Following (Yang et al., 2024; 2023), we select categories that may contain privacy-sensitive information, such as human faces, vehicle registration plates, and books. We compare the original client images with the synthetic images. The visualization results are shown in Figure 5. The results indicate that the initial noise sampled from the fitted GMM still retains the diversity of the DM generation process. As a result, the synthetic dataset only preserves the styles of the client data and does not directly reconstruct the original client images or any privacy-sensitive content within them, thereby preventing privacy leakage. However, we must acknowledge that, in theory, when the number of client images is extremely limited, there is a very small probability that

the fitted GMM could directly sample a noised original client image. This could potentially result in some degree of privacy leakage. Moreover, accurately quantifying the potential risk of privacy leakage remains a major challenge in OSFL methods based on DMs, and this will be a key focus of our future work.

