# OpenReview forum: "Local Distribution-Conditioned Image Synthesis for One-Shot Federated Learning"
_ICLR.cc/2026/Conference — Submitted to ICLR 2026_

### Official Review · Reviewer_vPzd · 2025-10-14

**Soundness:** 3
**Presentation:** 3
**Contribution:** 2
**Rating:** 4
**Confidence:** 3

**Summary:**

This paper proposes a method for One-Shot Federated Learning (OSFL), called FedLD. FedLD fits a Gaussian Mixture Model (GMM) to the noised latent representations of client’s images, which are then uploaded to the server for image generation. The GMM is used to generate initial noise which are then denoised to produce the training images for the target classification model. Experiments on multiple benchmarks show the promising results of FedLD.

**Strengths:**

1. The writting is clear and the proposed method is easy to follow.
2. The proposed method of using diffusion models to improve communication/training efficiency is interesting.
3. The proposed method achieves promising results among multiple benchmarks.

**Weaknesses:**

1. The paper Fedbip [1] seems to apply a similar idea of transmitting the noised image vectors between clients and server compared with the proposed method. However, the authors do not compare with this work. There should be either empirical comparison by experiments or a dicussion regarding the differences of the two methods.
2. The proposed method outperforms the ceiling performance in Table 1, which is unexpected. The authors should further discuss this.
3. The diffusion models are good at generating natural images, how does the method performs when generating rare domain images, e.g., medical images? (which is also considered as a critical application for Federated Learning)
4. What is the number of Gaussian components (K) used in the experiments?

[1] Chen, Haokun, et al. "Fedbip: Heterogeneous one-shot federated learning with personalized latent diffusion models." Proceedings of the Computer Vision and Pattern Recognition Conference. 2025.

**Questions:**

Please refer to the Weaknesses section.

---

### Official Review · Reviewer_6gPL · 2025-10-20

**Soundness:** 3
**Presentation:** 3
**Contribution:** 3
**Rating:** 6
**Confidence:** 2

**Summary:**

The paper proposes a one-shot federated learning (OSFL) method that encodes local data distribution into the diffusion model’s initial noise. Each client fits a GMM on its noised latent features and uploads only the GMM statistics; the server then samples initial noise from these GMMs and, combined with class text prompts, generates synthetic data to train a global classifier.

**Strengths:**

The idea of encoding client-specific styles/backgrounds in the diffusion model’s initial noise is simple yet effective, and the experimental design is reasonably comprehensive.

**Weaknesses:**

1.The paper lacks an evaluation of how accurately the Gaussian Mixture Model (GMM) fits the data. It would be helpful to include quantitative analyses comparing the fitted GMM distribution to the true noise (or latent) distribution — for example, through negative log-likelihood, KL divergence, or visual/statistical comparison between sampled and real noise. Such evaluation would strengthen confidence in the proposed representation quality.

2.The paper could benefit from citing or discussing literature showing that diffusion models tend to preserve similar styles after denoising noisy inputs. This part is currently missing in the motivation and related work.  Adding prior studies or experiments demonstrating that “adding noise and then denoising retains style information” would help validate the effectiveness of the proposed mechanism.

**Questions:**

1. **Effect of the diffusion step (T) / noise strength.**
  * How does the chosen (T) affect (a) the GMM fit on noised latents and (b) the final generation quality?
 * Do all clients use the same (T)? If not, does heterogeneity in (T) introduce bias or mismatch between client GMMs and server-side generation? A sensitivity study over (T) would clarify robustness.
2.   Would it be possible to report Local-only (train and evaluate on each client’s own data) and/or Local + Generated (train locally with the client’s own real data plus its generated data) as reference lines—particularly in the Unique NICO++ setting?
3.  In Unique NICO++, what happens if a client’s GMM noise (carrying its background/style) is combined with class prompts that do not appear on that client? Does the method still help the global model on those unseen classes, or does it harm class semantics?
4.  Results appear sensitive to the number of generated images per prompt (R).
   * For every setting, what is the total number of synthetic samples used by your method and by each baseline? Are the comparisons data-budget matched (per class/client and in total)?
   * Have you explored the upper bound scaling (e.g., larger (R)) to characterize marginal gains vs. compute? Even a partial scaling curve would be informative.

---

### Official Review · Reviewer_jKS7 · 2025-11-04

**Soundness:** 1
**Presentation:** 1
**Contribution:** 1
**Rating:** 2
**Confidence:** 4

**Summary:**

The paper proposes FedLD for one-shot federated learning (OSFL). Each client encodes its local images with the server’s Stable Diffusion encoder, adds forward-process noise to those latents, fits a Gaussian Mixture Model (GMM) to the “noised-latent” distribution, and uploads only the GMM parameters. The server then samples initial latents x_T from each client’s GMM and runs a pre-trained diffusion model (with class text prompts) to synthesize per-client datasets; a single global classifier is trained on these synthetic images. Experiments on DomainNet, OpenImages, and NICO++ include visualizations and comparisons against multi-round FL baselines and several diffusion-based OSFL variants.

**Strengths:**

- Clear, simple communication pattern for OSFL: clients upload small GMM statistics rather than models or features.
- Empirical visuals make the idea intuitive (domain/style control through initial noise).
- Communication/computation accounting is attempted (tables on upload/download and GFLOPs).
- The dataset partition schemes attempt to cover both feature and label heterogeneity, with explicit (if strong) non-overlap assumptions.

**Weaknesses:**

Novelty and positioning
- Method is not novel in a substantive ICLR sense. Fitting per-client GMMs and sampling x_T to steer a pre-trained diffusion model is a straightforward combination of standard components; there is no new learning principle, estimator, or algorithm with provable properties.
- The work argues initial-noise conditioning is the key insight, but closely related conditional or prior-based controls already exist; the paper does not articulate what is fundamentally new beyond “use a GMM for initial latents.”
- The paper compares to some diffusion-OSFL baselines, but the broader FL/KD literature and alternative lightweight client summaries are under-covered, weakening claims of originality.

Objective and conceptual clarity
- The formal objective (Eq. 1) is a standard global FL objective over a single aggregated model F_g, yet the text frequently speaks about “adapting to each client distribution.” There is no explicit personalization step, so the target seems global—even as the pipeline is per-client conditional during data synthesis. This mismatch should be clarified and reflected in metrics.
- The description of Stable Diffusion is inaccurate: the paper says “Stable Diffusion first maps the initial noise into the latent space via E.” In SD, E encodes images to latents; initial noise is sampled directly in latent space. This is not a minor wording issue because the client-side “distribution extraction” depends on how latents/noise are conceptualized.
- The link from a GMM over highly noised latents to downstream class-conditional image fidelity is argued mostly by intuition and visuals; no analysis of when/why initial-noise matching preserves client style/background while maintaining class semantics.

Privacy and security
- Privacy claims are weak. Uploading per-client GMM means and covariances of noised latents can leak information about client distributions; no formal threat model, differential privacy, secure aggregation, or attack evaluation (membership/attribute inference, reconstruction) is provided. The paper itself acknowledges a non-zero chance of sampling near-original content and leaves quantification to future work. Calling this “strong privacy protection” is not supported by evidence.

Experimental rigor and fairness
- The main results claim to “often outperform” centralized training (the “Ceiling”). But FedLD trains on images synthesized by a powerful, web-scale pre-trained diffusion model with class prompts—i.e., it injects external prior knowledge the centralized baseline does not use. This is not an apples-to-apples “upper bound” comparison; it demonstrates the benefit of external generative priors, not superiority of OSFL per se.
- The multi-round FL baselines are capped at 20 rounds, which is a modest budget for heterogeneous, medium-resolution vision and may under-serve strong optimizers; tuning parity and convergence diagnostics are not shown.
- Heterogeneity choices are specialized (strict domain non-overlap). While interesting, this setting advantages style-transfer-like synthesis; results under other realistic partitions (e.g., Dirichlet label skew with partial domain overlap) are missing.
- Communication/computation accounting mixes one-shot costs for FedLD with multi-round totals for FedAvg-style methods; units are inconsistent across tables (MB vs GB), and it is unclear whether all one-time costs (e.g., distributing the SD encoder) are counted equivalently.
- Ablations mainly vary the number of generated images (R) and choice of diffusion model; they do not probe the core design (e.g., effect of GMM component count, covariance tying vs full, robustness to GMM misfit, or the necessity of per-client GMM vs a global prior).
- Reporting lacks uncertainty estimates (CIs, seed spread) for many tables; statistical significance is not established.

Mathematical/notation issues
- Notation around the forward process uses a single \(\bar a_T\) constant; schedules typically depend on t with \(\bar\alpha_T\). If the intent is to simulate x_T, say so precisely; otherwise the definitions are ambiguous.
- Equation (8) blends variables/constants from different sampler variants without a self-contained derivation or references; as written it is hard to verify correctness.

Presentation
- Figure 1 helps, but key roles (what is encoded where, what is sampled where) could be clearer; the “Local Distribution Fitting” box implies uploading GMM stats but not the encoder-distribution step.
- Minor language and caption issues (e.g., “Visualization pf privacy-sensitive…”). Proofreading needed.

**Questions:**

1) Is the target global FL or personalized FL? If global, why design per-client synthetic datasets instead of directly sampling from a mixture of client GMMs into one training set; if personalized, where is the personalization step in optimization/metrics?
2) How exactly does a GMM fitted to heavily noised latents preserve client-specific style/background after reverse diffusion? Can you analyze the sensitivity to the noise level and to the number/shape (full vs diagonal) of mixture components?
3) Please fix the Stable Diffusion description (encoder role) and clarify the client-side pipeline mathematically.
4) Provide a formal privacy threat model and at least basic attack evaluations (membership/attribute inference) against uploaded GMMs and generated images; consider DP on uploaded statistics and assess the utility-privacy trade-off.
5) Revisit the “Ceiling” comparison: train a centralized model that is allowed to augment with the same diffusion prior and prompts, or avoid calling centralized supervised training an “upper bound.”
6) Add core ablations on the proposed mechanism: (i) vary mixture count and covariance tying, (ii) replace per-client GMM with a single global GMM, (iii) replace GMM with simple Gaussian, (iv) use text-only prompts vs prompts + GMM, to isolate where gains come from.
7) Report seeds and confidence intervals; expand training budgets for the multi-round baselines and show convergence curves; strengthen heterogeneity settings beyond strict domain non-overlap.

---

### Meta-Review · Area_Chair_TJAJ · 2026-01-04

**Summary:**

The method has limited novelty and largely combines existing components without a new learning principle or theory. The paper is conceptually unclear about whether it targets a global or personalized federated objective. Privacy claims are not supported by a formal threat model or empirical evaluation. Experimental comparisons are confounded by strong diffusion priors and insufficiently controlled baselines, and the core mechanism lacks mechanistic justification.

**Reviewer Concerns:**

There was no rebuttal submitted. As a result, it is reasonable to assume reviewers won't change their scores.

**Reviewer Scores:**

Unclear, since there was no rebuttal response from the authors.

---

### Decision · Program_Chairs · 2026-01-26

Reject